# Cytokinin Promotes Jasmonic Acid Accumulation in the Control of Maize Leaf Growth

**DOI:** 10.3390/plants12163014

**Published:** 2023-08-21

**Authors:** Aimee N. Uyehara, Angel R. Del Valle-Echevarria, Charles T. Hunter, Hilde Nelissen, Kirin Demuynck, James F. Cahill, Zachary Gorman, Georg Jander, Michael G. Muszynski

**Affiliations:** 1Department of Tropical Plant and Soil Sciences, University of Hawaiʻi at Mānoa, Honolulu, HI 96822, USA; 2Chemistry Research, Center for Medical, Agricultural and Veterinary Entomology, USDA-ARS, Gainesville, FL 32608, USA; charles.hunter@usda.gov (C.T.H.);; 3Department of Plant Biotechnology and Bioinformatics, Ghent University, 9052 Gent, Belgium; 4Center for Plant Systems Biology, VIB, 9052 Gent, Belgium; 5Department of Genetics, Development and Cell Biology, Iowa State University, Ames, IA 50011, USA; 6Boyce Thompson Institute, Cornell University, Ithaca, NY 14853, USA

**Keywords:** jasmonic acid, maize, cytokinin

## Abstract

Plant organ growth results from the combined activity of cell division and cell expansion. The co-ordination of these two processes depends on the interplay between multiple hormones that determine the final organ size. Using the semidominant *Hairy Sheath Frayed1* (*Hsf1*) maize mutant that hypersignals the perception of cytokinin (CK), we show that CK can reduce leaf size and growth rate by decreasing cell division. Linked to CK hypersignaling, the *Hsf1* mutant has an increased jasmonic acid (JA) content, a hormone that can inhibit cell division. The treatment of wild-type seedlings with exogenous JA reduces maize leaf size and growth rate, while JA-deficient maize mutants have increased leaf size and growth rate. Expression analysis revealed the increased transcript accumulation of several JA pathway genes in the *Hsf1* leaf growth zone. A transient treatment of growing wild-type maize shoots with exogenous CK also induced the expression of JA biosynthetic genes, although this effect was blocked by the co-treatment with cycloheximide. Together, our results suggest that CK can promote JA accumulation, possibly through the increased expression of specific JA pathway genes.

## 1. Introduction

Growing plants accumulate biomass over time through the integration of cell division and cell expansion. These processes produce biomass by increasing the cell number (cell division) and increasing the final cell volume (expansion). In many model plants such as *Arabidopsis* or maize, leaf growth follows a basipetal pattern where differentiation starts at the distal tip of the leaf and finishes near the proximal base [1,2]. In grass leaves, the basipetal growth mechanism sets up regions of cell division, elongation, and maturation that are linearly organized and spatially separated into distinct growth zones [3]. Elucidating the molecular processes that control the specification and size of these growth zones is key to improving crop growth.

Plant hormones are molecular messengers with low molecular weights that regulate growth, development, and defense [4,5,6,7]. Generally, plant hormones can be divided into two classes: growth hormones and defense hormones. Classical growth hormones include cytokinin (CK), gibberellins (GAs), brassinosteroids (BRs), and auxin [5]. These hormones have been ascribed functions in cell proliferation, stem elongation, seed germination, and organ elongation, respectively. Classical defense hormones include salicylic acid (SA), jasmonic acid (JA), and ethylene (ET), and are responsible for the majority of signaling in response to pests and pathogens [5]. In addition to their primary functions as growth or defense hormones, crosstalk occurs between the growth and defense hormone signaling pathways to manage resources between those two processes [8,9]. This interplay results in a growth–defense tradeoff. One described example of crosstalk is the signaling between the growth hormone GA and the defense hormone JA. In the presence of JA, the GA repressor DELLA is released to bind and degrade GA, leading to suppression of GA-mediated growth by JA [10]. In contrast, BR seems to relieve JA-induced growth suppression, suggesting an antagonistic relationship between BR and JA [11,12]. Crosstalk has also been shown to occur between the defense hormone SA and auxin, wherein SA represses auxin-mediated growth by repressing the transcription of the F-box protein TIR1/AFB, and, thus, stabilizing the auxin repressor AUX/IAA [13,14]. As predicted by the growth–defense tradeoff model, signaling by defense hormones is antagonistic to growth hormones and typically leads to growth suppression.

Cytokinin (CK) is a growth-promoting hormone that regulates shoot growth, apical dominance, senescence, and the promotion of cell proliferation [15,16]. In dicots, cytokinin promotes leaf growth by stimulating cell division. This has been demonstrated through exogenous CK treatment, the overexpression of CK catabolic enzymes, or the knockout of CK receptors. For example, decreasing the endogenous CK concentration through the overexpression of the CK catabolic enzyme, CYTOKININ OXIDASE (CKX) in *Nicotiana tabacum,* reduced the leaf size by reducing the cell number [16]. In *Arabidopsis thaliana*, the reduction of CK signaling through the knockout of the CK receptors *Arabidopsis* HISTIDINE KINASE 2 (AHK2), AHK3, and CRE1/AHK4 resulted in plants with a severely reduced rosette size and a reduced number of cells per leaf [17]. A reduced cell number as a result of reduced cytokinin perception or signaling resulted in growth compensation through cell expansion [16,17]. In contrast, constitutively active CK receptor mutants in *A. thaliana* exhibited larger leaves with more epidermal cells due to either an extended period of mitotic activity, increased mitotic rate, or both [18]. From these examples, increased CK signaling usually promotes leaf growth, and decreased CK signaling typically reduces leaf growth.

The role of CK in regulating monocot leaf growth is less defined. Loss-of-function mutations in two of the four rice CHASE-domain histidine kinase receptors produced reduced shoot and root growth phenotypes, consistent with the effects seen in *Arabidopsis* [19]. In contrast to *Arabidopsis* and rice, maize has seven CHASE-domain histidine kinase receptors [20,21]. The only monocot CK receptor gain-of-function mutant is the semidominant maize mutant *Hairy Sheath Frayed1* (*Hsf1*) that resulted from EMS-induced missense mutations in the cytokinin receptor, *Zea mays HISTIDINE KINASE1* (*ZmHK1*), an orthologue of *AtHK4* [22,23]. Although CK typically promotes cell division and growth, the increased signaling (hypersignaling) of CK in *Hsf1* mutants caused reduced leaf growth compared to wild-type siblings [23]. The characterization of *Hairy Sheath Frayed1* (*Hsf1*) demonstrated the role the increased CK signaling had on leaf patterning, leaf size, and epidermal cell fate [22,23]. The effect of the reduced CK content on monocot growth was indirectly observed through the transgenic overexpression of zeatin O-glucosylzeatin (ZOG), an enzyme that inactivates and sequesters CK through the addition of a sugar moiety [24]. Homozygous *Ubi:ZOG1* maize lines showed CK deficiency phenotypes such as reduced growth and, interestingly, a feminized tassel [24].

Jasmonic acid (JA) is an established plant growth regulator involved in processes such as leaf senescence, plant defense, and male fertility [25]. Linolenate lipoxygenase (LOX) catalyzes the first step of JA biosynthesis from chloroplast membrane phospholipids [26]. The resulting hydroperoxyoctadecadienoic acids are further converted into (+)-7-iso-JA via allene oxide synthase (AOS), allene oxide cyclase (AOC), 12-oxophytodienoic reductase (OPR), and three cycles of ß-oxidation [26]. Bioactive JA-Ile is formed through the conjugation of isoleucine by the jasmonate amido synthetase (JAR) [26]. The catabolism of JA-Ile occurs through the hydroxylation of JA-Ile to 12-hydroxy-JA-Ile (12OH-JA-Ile) and the subsequent oxidation to 12-carboxy-JA-Ile (12COOH-JA-Ile) by the cytochrome CYP94B and CYP94C enzymes, respectively [27,28,29,30,31,32]. 12OH-JA-Ile is a bioactive JA that can trigger JA signaling and induce similar growth responses as JA-Ile [31,33,34]. Understanding JA’s role in both defense signaling and in regulating plant growth is aided by studies on biosynthesis and signaling mutants. In maize, mutants for LOX, OPR, and CYP94B include *tasselseed1* (*ts1*), *opr7-5 opr8-2*, and *Tasselseed5* (*Ts5*), respectively [32,35,36]. These mutants add to a growing body of research that establishes JA as a growth repressor. Initial studies showed that exogenous JA application to rice seedlings reduced seedling leaf size [37]. Additionally, the wound induction of JA and analysis of *Arabidopsis* JA biosynthesis mutants have shown that JA suppresses cell proliferation, leading to smaller leaves with fewer and smaller epidermal cells [38,39].

The linear organization of the maize leaf growth zones makes it straightforward to use kinematic analysis to measure the relative contributions of division and expansion to final leaf size [40]. Leaf growth zone analysis also provides insights into the complex molecular interactions underlying leaf growth as different hormones have measurable and distinct impacts on the different growth zones. This was demonstrated through the analysis of maize GA biosynthesis mutants, where a higher bioactive GA content increased the size of the division zone and determined the spatial location of the transition between the division and elongation zones [41]. These data also implicated other growth hormones such as cytokinin, auxin, and brassinosteroids as possible contributors to division zone size [41].

Here, we show that increased CK signaling reduces cell division in the leaf growth zone through the promotion of JA accumulation. To do this, we used exogenous hormone treatments, hormone biosynthesis and signaling mutants, the kinematic analysis of leaf growth, and expression analysis. Altogether, our data identified a previously unrecognized connection between cytokinin and the defense hormone JA in regulating maize leaf growth.

## 2. Results

### 2.1. Hsf1 Mutants Have a Reduced Leaf Growth Phenotype

We have previously shown that the heterozygous, semidominant *Hsf1*/+ mutants have smaller leaves and that exogenous CK treatment can phenocopy this effect [23] (Figure 1A, Appendix A). To further characterize this reduced growth phenotype, the leaf size, growth rate, and leaf elongation duration of seedling leaf #4 was determined for the *Hsf1*/+ and wild-type sibling plants in three different genetic backgrounds (Figure 1A,B, Appendix A). In all three backgrounds, the *Hsf1*/+ leaf #4 blade length was reduced by 10–20% compared to their wild-type siblings (Figure 1A, Appendix A). Consistent with a reduced blade size, the leaf elongation rate (LER) was also reduced by 20–25% across the three backgrounds (Figure 1B, Appendix A). Interestingly, leaf elongation duration (LED) was slightly increased for the *Hsf1*/+ leaf #4, which may account for the fact that the reduction in leaf size is not as great as the reduction in LER would predict. To determine the cellular basis underlying this growth rate reduction, kinematic analysis was performed on the *Hsf1*/+ and wild-type siblings in the B73 genetic background. The kinematic analysis showed that *Hsf1*/+ mutants had fewer dividing cells in the division zone and, thus, had a smaller division zone in leaf #4 compared to wild-type (Figure 1C). These data suggested that CK hypersignaling in *Hsf1*/+ mutants reduced cell divisions in the leaf growth zone, which slowed the growth rate, resulting in a smaller leaf.

### 2.2. Distinct Jasmonates Accumulate in Growing Hsf1/+ Maize Leaves

To determine if CK hypersignaling in the *Hsf1* mutant was affecting other hormones that may impact growth, the phytohormone content was measured in WT and *Hsf1/+* whole seedlings. The *Hsf1*/+ seedlings accumulated 2-fold more 12COOH-JA-Ile, 1.5-fold more 12OH-JA, and 1.3-fold more 12OH-JA-Ile compared to wild-type (Figure 1D). A few other hormones showed modest accumulation differences but not in a pattern consistent with the *Hsf1* reduced growth phenotype.

To obtain the spatial resolution of the elevated JA content in *Hsf1*, seedling leaf #9 was sampled at steady-state growth, divided into thirds along the proximal–distal axis, and the JA content was determined. Consistent with the whole seedling data, the JA content was elevated two- to three-fold across the entire *Hsf1*/+ leaf, including the growth zone (Figure 1E). Although CK had not previously been shown to affect the JA content, JA is known to inhibit cell division in eudicots, providing a possible mechanism by which the *Hsf1* mutation conditioned reduced growth [37,38,39].

### 2.3. JA Pathway Genes Are Upregulated in the Leaf Growth Zone of Hsf1 Mutants

Given that the JA content was increased in *Hsf1* mutants, we assessed whether the expression of JA pathway genes was increased in the *Hsf1* leaf growth zone. The growth zone of leaf #4 at steady-state growth was partitioned into 5 mm subsections providing a high-resolution spatial sampling through the division, first transition, and elongation zones (Figure 2). The subsections were collected in triplicate and transcript levels for select JA pathway genes were measured by a quantitative real-time PCR. These genes were chosen to survey key steps in JA biosynthesis and because mutants are available for some [25,42] (Figure 2). We found the JA biosynthetic genes *ts1*, *ZmAOC2,* and *opr7* were significantly upregulated in the division zone of *Hsf1/+* leaves (Figure 2). This suggested that the increased JA accumulation was due to the increased expression of at least some JA biosynthetic genes in the division zone of *Hsf1* mutant leaves. In addition, the JA-responsive gene *ZmMYC7,* an orthologue of AtMYC2, also had higher expression early in the growth zone in *Hsf1/+*, suggesting increased JA levels were being perceived by the JA signaling pathway (Figure 2). Overall, the expression data supports the hypothesis that CK signaling promotes JA accumulation through the increased expression of JA biosynthetic genes.

### 2.4. Exogenous Jasmonic Acid Treatments Reduce Leaf Growth Rate in Maize

To test if increased expression of JA biosynthetic genes could be responsible for reduced leaf growth in the *Hsf1* mutant, B73 inbred maize seeds were transiently treated with 1 mM JA and effects on seedling leaf growth were assessed (see Section 5 for details). Exogenous JA treatment of germinating maize seeds resulted in a 25–30% reduction in sheath and blade length for seedling leaves #1 to #4 (Figure 3A, Appendix A). JA treatment also promoted reductions in blade width which varied between 9–20% depending on leaf number (Figure 3A and Appendix A). Similar to the effects of JA in other plant systems, these data indicated that JA treatment can reduce leaf size in maize seedlings.

The JA mediated decrease in leaf size could have resulted from a reduction in growth rate, the duration of growth, or both. To distinguish between these possibilities, the LER and LED were determined for leaf #4 from B73 seedlings treated with different concentrations of JA. While both the control and JA-treated plants maintained steady-state growth for five days, the 1 mM JA-treated seedlings had a pronounced reduction in LER compared to control throughout the period of steady-state growth (Figure 3B). Lower concentrations of JA did not affect LER (Appendix A) and no change in LED was observed for any JA concentration tested. To determine the minimum time of JA treatment required to elicit growth reduction, germinating B73 seeds were treated with 1 mM JA for 1, 6, 12, 24, and 48 h (see Section 5 for details). Decreased blade length and width were observed only after 48 h of JA exposure for leaves #1 to #3 (Appendix A). Thus, exogenous JA treatment for at least 48 h could decrease maize leaf size by reducing the growth rate.

### 2.5. Hsf1 Is Less Responsive to Exogenous Jasmonic Acid Treatment

Because *Hsf1* mutant leaves have more JA and are smaller than wild-type, we hypothesized that the leaf size of *Hsf1* mutants would be less responsive to exogenous JA treatment than wild-type siblings or the B73 inbred. To test this, we treated germinating seeds that were segregating 50% *Hsf1*/+ and 50% wild-type with 1 mM JA. The excessive pubescence *Hsf1* phenotype (increased macrohair density on the abaxial sheath) was not affected by exogenous JA treatments and was used to score seedlings as either *Hsf1*/+ or wild-type. As expected from previous analysis, the leaf size in untreated *Hsf1*/+ was reduced by approximately 20% compared to untreated wild-type siblings (Figure 4A, Appendix A). JA treatment reduced the leaf size in both wild-type and *Hsf1/+* genotypes compared to their respective controls (Figure 4A). However, the response to JA in *Hsf1/+* plants was not as great as in wild-type plants, as leaf size reduction was dependent on the leaf tissue and parameter measured. JA treatment reduced the wild-type sheath length, blade length, and blade width by about 15–25%, similar to reductions seen in JA-treated B73 seed, although the blade #4 width was not affected (Figure 4A and Appendix A). In contrast, only the blade length was consistently reduced (17–25%) in JA-treated *Hsf1*/+ plants, with no significant reduction in sheath length or blade width (Figure 4A and Appendix A). These results suggest that, in the *Hsf1/+* mutant, the blade length but not the other leaf growth parameters are responsive to the JA treatment.

Since JA treatment further reduced *Hsf1/+* blade size, we sought to determine whether the JA treatment was affecting the growth rate or duration of growth. To do this, LER and LED were determined for leaf #4 of the seedlings from the 1 mM JA-treated 1:1 segregating *Hsf1* and wild-type seeds (as above). As seen previously, compared to the untreated wild-type siblings, the untreated *Hsf1*/+ had a reduced LER and extended LED (Figure 4C). Also similar to our results with JA-treated B73, JA-treated wild-type LED was not affected but LER was reduced, which was especially evident in the first 2.5 days of growth (Figure 4D). In contrast, *Hsf1/+* LER, especially during the first 2.5 days of steady-state growth, was not affected by JA treatment. Instead, LED was reduced by JA treatment in *Hsf1*/+ plants where steady-state growth began to slow starting at 3 days, instead of 5 days, and continued to slow until leaf growth stopped by day 8 (Figure 4E). Finally, JA-treated wild-type and JA-treated *Hsf1*/+ growth showed similar growth patterns to what was seen for these genotypes without JA treatment (Figure 4F). Thus, although *Hsf1*/+ blade length can be reduced further by JA treatment, it is likely caused by a shortened LED, since LER was not impacted. This can be seen when comparing the actual leaf length (sheath length + blade length) of growing leaf #4 from both genotypes with and without JA treatment (Figure 4C–F). The leaf length was reduced at each time point during leaf growth for wild-type vs. *Hsf1*/+, for wild-type vs. JA-treated wild-type, and for JA-treated wild-type vs. JA-treated *Hsf1*/+ (Figure 4C,D,F). In contrast, *Hsf1*/+ vs. JA-treated *Hsf1*/+ showed the leaf length was not different until after 7 days of leaf growth, nearly the time growth stopped (Figure 3E). This suggests that, in *Hsf1/+* mutants, where steady-state leaf growth is reduced, possibly by increased JA content, additional JA can further reduce leaf size by truncating the duration of growth.

### 2.6. Growth Is Enhanced in Jasmonic Acid-Deficient Mutants

Our data are consistent with previous work showing JA can reduce growth. This implies that reduced endogenous JA accumulation may enhance growth, leading to larger leaves. To understand how endogenous concentrations of JA might affect leaf growth and size, we measured the leaf size and growth in a number of JA-deficient maize mutants [32,36]. Duplicate genes encode 12-OXO-PHYTODIENOIC ACID REDUCTASE (OPR), a key enzyme in the JA biosynthetic pathway responsible for converting OPDA into (+)-7-iso-JA, which is later modified into bioactive JA [36]. Plants homozygous for recessive null mutations in both the *opr7* and *opr8* genes are JA-deficient, display a feminized tassel or “tasselseed” phenotype, and have longer seedling leaves #1 and #2 [36]. A single functional *opr* allele at either locus is sufficient for maintaining wild-type levels of JA content and plant phenotype. Using a population that was homozygous null for the *opr7-5* allele and segregating for wild-type and null *opr8-2* alleles, we assessed the leaf size and leaf growth in JA-sufficient and JA-deficient genotypes (Figure 5A,B). As was shown previously, the leaf #1 and #2 sheath and blade lengths of the JA-deficient genotype were increased by 20%–48%, and the leaf #3 and #4 blade length were increased by 13%–24% (Figure 5A,B and Appendix A). Interestingly, the sheath length was increased for leaf #3 but decreased in leaf #4 in the *opr7 opr8* double mutant (Appendix A). We also noted that the blade width increased in leaf #3 and #4 by 9%–18% in the JA-deficient genotype. Overall, in *opr7 opr8* double mutants, increases in sheath and blade length diminished from leaf #1 to #4 but the blade width was smaller than wild-type in leaf #1 but larger than wild-type in leaf #4. An assessment of the growth rate in the double *opr7 opr8* mutant revealed an increase in LER and LED compared to the JA-sufficient genotypes (Figure 5B). This suggested the lack of JA increased both the rate and the duration of leaf growth.

To extend the results above, we also measured the leaf size and growth in the semidominant, gain-of-function *Tasselseed5* (*Ts5*) mutation [32]. The *Ts5* locus encodes a cytochrome P450 enzyme, ZmCYP94B1, that oxidizes the bioactive JA-Ile to 12OH-JA-Ile which is less bioactive, and *Ts5* mutants express more *ZmCYP94B1* than wild-type [32]. Thus, *Ts5*/+ plants have a lower JA content than wild-type siblings and display the tasselseed phenotype expected for JA-deficient mutants. *Ts5/+* was crossed to *Hsf1*/+ and the 1:1:1:1 segregating population was analyzed for LER and LED. LER and LED was measured and plants were genotyped for *Ts5*/+. First, we analyzed *Ts5/*+ growth compared to wild-type. *Ts5*/+ plants exhibited an increased LER compared to wild-type and possibly an increase in growth duration (Figure 5C). Consistent with the results from the *opr7 opr8* population, these JA-deficient mutants showed increased growth rate, supporting the role of the reduced JA promoting leaf growth.

### 2.7. JA-Deficient Mutants Suppress the Reduced Leaf Growth Phenotype in Hsf1 Mutants

Using the population described in Figure 5C, we next compared the LER and LED of single and double mutants. *Hsf1*/+ mutants had reduced LER and an extended LED compared to wild-type, as seen from the previous characterization of *Hsf1*/+ growth (Figure 1B, Figure 1). *Ts5*/+, as stated in Figure 5C, had increased LER compared to WT. Interestingly, the average LER for the double mutant *Hsf1/+ Ts5*/+ closely matched the wild-type, except for at the 48 hr time point, where WT LER slightly exceeded the *Hsf1*/+ *Ts5*/+ LER (Figure 6A). An analysis of the final leaf lengths of the entire population showed that combining *Hsf1*/+ and *Ts5*/+ resulted in a final leaf length similar to wild-type plants (Figure 6B).

### 2.8. Exogenous CK Treatment Induces Expression of JA Pathway Genes in the Leaf Growth Zone

Since the expression of several JA pathway genes was higher in the leaf growth zone of the *Hsf1* CK hypersignaling mutant, we asked if exogenous CK application to maize inbred seedlings could also induce JA pathway gene expression in the leaf growth zone. To do this, 10-day-old B73 seedlings were cut at the root–shoot junction, and shoots were incubated for 1, 2, and 4 h with 10 µM 6-BAP (details in Section 5). After incubation, the basal 2 cm of leaf #4, encompassing the division zone and part of the expansion zone, was collected, and gene expression was quantified using qRT-PCR. We first determined that the exogenous CK application was perceived by assessing the expression of three CK early-response genes: the type A response regulators *ZmRR3* and *ZmRR6*, and *cytokinin oxidase2* (*ckx2*). Type A response regulators are negative regulators of CK signaling that are rapidly expressed without de novo protein synthesis upon CK treatment [43,44]. As expected, *ZmRR6* transcripts were upregulated in the growth zone by 1 h, and all three CK reporters showed robust expression by 4 h (Figure 7A). Thus, the growth zone of leaf #4 was perceiving and responding to the CK application by 4 h. We next assessed the JA pathway expression in these same tissues. Of the genes surveyed, we found an increase in the expression of both the JA biosynthesis and catabolism genes. Specifically, *ts1*, *aos1a*, *aos2a*, *aoc2*, *opr7*, and *Ts5* all showed a 1.5- to 3-fold increase in expression after 4 h of CK treatment. This showed that CK could induce JA pathway gene expression in the growth zone after 4 h.

We next examined whether the CK-induced increase in JA gene expression required new protein synthesis downstream of CK signaling. We considered two possibilities: (1) the CK treatment and subsequent signaling resulted in the downstream phosphorylation and activation of a transcription factor, such as a type-B response regulator, or (2) the CK treatment and signaling resulted in the transcription and translation of a new transcription factor that activated the expression of the upregulated JA genes. To test this, the CK application on cut B73 seedling shoots was performed with and without cyclohexamide (CHX), a translational blocker. We hypothesized that if the CK-induced expression of JA genes was dependent on de novo protein synthesis, a combined treatment with CK and CHX would result in no increased expression of JA-pathway genes. However, if JA genes were directly regulated by CK-signaling components, like the expression of *ZmRR3* and *ZmRR6*, JA gene expression would still be increased in the combined CK- and CHX-treated samples. We also tested whether the combined CK and CHX treatment would work as predicted by assessing the expression of the three CK reporters. As expected, since the type-A response regulator expression does not require de novo protein synthesis, *ZmRR3* and *ZmRR6* expression increased in the combined CK and CHX treatment, although the increase was smaller than with CK alone (Figure 6B). In contrast, the CK-induced increased expression of JA pathway genes was abolished with CHX treatment (Figure 6B). This suggests that CK induces the transcription and translation of a new protein that regulates the JA biosynthesis gene expression in the leaf growth zone. Our CK-induction system will be useful in identifying the CK-induced regulators of these JA pathway genes.

## 3. Discussion

Many dicot examples show that CK signaling promotes the accumulation of plant biomass [16,17,18]. However, results from the *Hsf1* mutant show that CK signaling in maize results in reduced shoot growth [23] (Figure 1A,B). This contrast may be integral to the differences between monocots and dicots. In contrast to the constitutive CK receptor mutant in *Arabidopsis*, which has larger leaves with more cells, *Hsf1* has smaller leaves due to a smaller division zone and reduced number of dividing cells [18] (Figure 1C). Due to the lack of CK-signaling mutants in monocots, it is difficult to tell if differential CK-mediated growth responses in monocots and dicots mark true differences in CK signaling or are due to absolute differences in endogenous CK concentrations and perception. However, rice *OsIPT3* transformants overexpressing the rate-limiting CK biosynthesis enzyme IPT3 resulted in stunted plants and provides another example of excess CK reducing plant growth [45].

To understand the connection between CK and leaf growth in *Hsf1*, we focused on characterizing the role of JA in regulating maize leaf growth because of the accumulation of hydroxylated JAs in *Hsf1* (Figure 1D,E) and the differential expression of JA biosynthesis genes in the division zone (Figure 2). Hydroxylated JAs in *Hsf1* may reflect changes in JA metabolism resulting from increased JA biosynthesis. Previous research has established that monocot and dicot growth is reduced through the JA-mediated inhibition of cell proliferation [25,37,38,39]. As expected, the exogenous application of JA to maize reduced LER, which, ultimately, reduced the leaf size (Figure 3A,B). Interestingly, the analysis of mutants deficient in JA (*opr7 opr8* and *Ts5*) show the increased final leaf size was due to increased LER and LED [25] (Figure 5). These data show that JA impacts growth primarily by decreasing LER, and support the role of JA-mediated growth reduction in *Hsf1* leaves.

Our data suggest that CK hypersignaling induces growth reduction in maize by crosstalk with the growth repressor JA (Figure 7). Crosstalk between CK and JA is not well-characterized and previous data linking the two have been indirect [46,47,48]. Most previous studies relied on exogenous treatments of CK and JA with mixed results that indicated a complex relationship between the two hormones [46,47,48]. One study found that JA treatment antagonized CK-mediated callus growth [48]. Our double mutant analysis of *Hsf1*/+ *Ts5*/+ reveal an antagonistic relationship between JA and CK, as the double mutant had wild-type LER and final leaf length (Figure 6A,B). In addition, we found that the CK treatment of B73 seedlings promotes the transcription and translation of an unidentified protein that promotes the expression of JA biosynthesis genes (Figure 7A,B). Further studies are needed to identify the CK-inducible regulators of the described JA genes.

The JA treatments of *Hsf1* suggest that CK crosstalk with other hormones, in addition to JA, may play a role in controlling *Hsf1* growth. While crossing the *Ts5*/+ with *Hsf1*/+ rescued the reduced growth phenotype of *Hsf1*/+, the *Hsf1*/+ growth pattern could not be phenocopied with the exogenous JA treatment (Figure 4F). This shows that the JA treatment reduces the wild-type leaf size to be equivalent with *Hsf1* (Figure 4A,B) and suggests that JA also reduces the leaf size by shortening the leaf elongation duration (Figure 4E). Differences between the *Hsf1*/+ *Ts5*/+ cross and the exogenous JA treatment of *Hsf1*/+ may stem from the strength of the JA perception or reveal the presence of another hormone that crosstalks with CK and JA. Specifically, the extended LED growth pattern is similar to that of a GA-signaling mutant, and provides another avenue of hormone crosstalk to investigate in the *Hsf1*/+ mutant [41]. Taken together, it is likely that JA is responsible for reducing LER, and another hormone controls LED in *Hsf1*.

## 4. Conclusions

In conclusion, our data suggest that CK hypersignaling upregulates JA biosynthesis genes, leading to growth reduction in the maize *Hsf1* leaf by suppressing cell proliferation. We provide evidence for an unidentified CK-inducible protein regulator that targets JA biosynthesis genes. Additionally, a growth analysis of JA-treated plants and JA-deficient mutants shows that JA impacts leaf growth by reducing LER, and the removal of JA promotes leaf growth by increasing LER. Collectively, these data highlight a new connection between CK and JA. Determining how CK connects to JA has the potential to provide new insights into the mechanisms plants use to balance growth and defense.

## 5. Materials and Methods

### 5.1. Plant Material, Genetics, Phenotypic Measurements, and Analysis

Inbred B73 was used as the standard maize line for all seed and seedling treatments. The CK hypersignaling mutant *Hsf1-1603* was previously described [23]. The JA-deficient *opr7-5 opr8-2* (we will refer to it as *opr7 opr8*) and *Tasselseed 5 (Ts5)* were previously described in Yan et al., 2012, and Lunde et al., 2005, respectively [32,36]. *Hsf1/+* plants were identified by the presence of macrohairs at the V1 stage and prongs in leaf margins past V6 [23]. JA-deficient mutants were grown in flats and genotyped by PCR using the primers described in Appendix A. Plants were crossed for several generations to produce the following genotypes to analyze: [*+/+, opr7, opr8/+*] WT, [*Hsf1/+, opr7, opr8/+*] CK-hypersignaling only, [+/+, *opr7, opr8*] JA-deficient only, and [*Hsf1/+, opr7, opr8*] CK-hypersignaling JA-deficient plants. In parallel, the following genotypes were developed: [+/+, *ts1/+*] WT, [*Hsf1/+, ts1/+*], CK-hypersignaling only, [+/+, *ts1*] JA-deficient, and [*Hsf1/+, ts1*] CK-hypersignaling JA-deficient plants. All genotypic classes were grown until leaf #4 matured.

### 5.2. Standard Germinating Seed Hormone Treatment

A stock and control solution of hormone was made as described by the manufacturer and stored at −80 °C. Surface-sterilized seeds imbibed overnight were placed embryo-face-down, about 20 seeds/Petri dish, onto a sterile paper towel and soaked with 2.5 mL of hormone at a working concentration (varied by hormone) in a 15 mm Petri dish. Typically, three biological replicates were carried out per treatment, using 20 seeds/Petri dish × 3 = 60 total seeds/treatment. The edges of the Petri dishes were sealed with parafilm to prevent evaporation and the entire Petri dish was wrapped in foil and placed in a lab drawer for six days. After six days of treatment, germinated seedlings were removed from the Petri dish, rinsed with sterile tap water, and transplanted to 1-gallon pots (Sunshine Mix #4 media, supplemented with 2 teaspoons osmocote, and 2 teaspoons ironite) and placed in the Pope greenhouse.

### 5.3. Cytokinin

First, 6-Benzylaminopurine (6-BAP) powder from Sigma Aldrich (St. Louis, MO, USA) was dissolved in 10 drops of 1 N NaOH, and brought to a concentration of 10 mM with sterile distilled water. A parallel water control stock was also made with 10 drops of 1 N NaOH. These stocks were further diluted to achieve the desired hormone treatment concentrations.

### 5.4. Jasmonic Acid

Then, 100 mg of JA (Sigma-Aldrich) was dissolved in 3 mL of 200-proof ethanol and 44.5 mL of sterile ddH_2_O to make a stock concentration of 10 mM JA. A control solution was made by adding 3 mL of 200-proof ethanol to 44.5 mL of ddH_2_O and stored at −80 °C. Both the JA and control solutions were diluted with sterile ddH_2_O until the desired working solution concentration was reached. Stock solutions were stored at −80 °C in 15 mL tubes. The working solution was made the day treatments started by diluting the 10 mM stock with sterile ddH_2_O to a final volume of 2.5 mL/Petri dish.

### 5.5. Final Leaf Size Measurements and Kinematic Analysis

Treated seedlings were grown until the fifth leaf was completely collared (the auricle and ligule that define the junction between the leaf sheath and blade were visible), ensuring that leaves #1 to #4 had completed growth. Sheath length, blade length, and blade width were measured for leaves #1 (most basal, first formed) to leaf #4. Leaves were measured by harvesting each leaf at its insertion into the stem. For sheath length, length was measured from the base of the sheath to the point at which the sheath transitions to the auricle at the midline of the leaf. For blade length, length was measured along the midrib from the auricle to the distal blade tip. For blade width, width was measured at the midpoint of blade length across the blade from margin to margin. Kinematic analysis of the *Hsf1/+* growth, elongation, and maturation zones was performed following Nelissen et al., 2013.

### 5.6. Growth Rate Measurement

Leaf elongation rates (LER) were taken when leaf #4 emerged from the whorl and was at steady-state growth, when LER is constant [49]. Briefly, the length of leaf #4 was measured as the distance from the insertion point of leaf #1 at the base of the plant to the tip of leaf #4 every 12 or 24 h until leaf #4 stopped growth (leaf length did not change for 2–3 consecutive time points). LER was calculated by dividing the difference in leaf length (cm) by the time elapsed (24 h). Leaf elongation duration (LED), the measure of time from when the leaf is 10 cm to final length, was determined from plotting LER by time elapsed. Leaf elongation duration (LED) was determined when steady-state growth stopped as observed when plotting LER by days post emergence of leaf #4 from the whorl. Finally, plants were dissected and leaf blade length, leaf blade width (measured at ½ the blade length mark), and leaf sheath length were measured on leaves #1–4.

### 5.7. Seedling Treatments and JA-Pathway Gene Expression Analysis

Seedling treatments were performed as described in [50] on B73 seedlings when leaf #4 was emerging from the whorl. Briefly, individual seedlings were cut at the shoot–root junction and submerged in 500 µL of 10 µM 6-BAP or equivalent control for 4 h. The basal 2 cm of the leaf, where division and expansion occur, was dissected and put in 500 µL of IBI Isolate (IBI Scientific; Dubuque, IA, CAT: IB47601) for RNA extraction following the manufacturer’s recommendations. RNA was quantified by using ND-1000 Spectrophotometer (Nanodrop, Wilmington, DE, USA). A total of 2 µg of RNA was used to synthesize cDNA with SuperScript IV VILO Master Mix with ezDNase Enzyme kit (Thermo Fisher Scientific; Waltham, USA, CAT: 11766050) following the manufacturer’s recommendations. Finally, 1:10 dilution of cDNA was used for RT- and quantitative RT-PCR.

Samples were initially screened for CK perception by RT-PCR amplifying *ZmRR3* (*abph1*; Zm00001d002982), a type-A response regulator that is only expressed when CK is present (Giulini et al., 2004), using the EconoTaq^®^ PLUS GREEN 2X Master Mix (Lucigen; Middleton, WI, USA) and following the manufacturer’s recommendations. The RT-PCR was performed using S1000™ Thermal Cyclers (Bio-Rad; Hercules, CA, USA) using the following cycling program: step 1 = 98 °C for 2 min, step 2 = 98 °C for 30 sec, step 3 = 60 °C for 30 s, step 4 = 72 °C for 30 s, step 5 = repeat steps 2–4 29 times, step 6 = 72 °C for 5 min, and step 7 = 10 °C. PCR products were run in 2% agarose gel electrophoresis using a 100 bp DNA ladder (GenScript; Piscataway, NJ, CAT: M102O).

Once perception was confirmed, genes that encode for the biosynthetic enzymes along the JA pathway were evaluated by quantitative RT-PCR using the iQ SYBR Green Supermix (Bio-Rad; CAT: 1708882) reagents, following manufacturer recommendations, and Bio-Rad CFX96 Touch™ thermocycler (Bio-Rad; Hercules, CA, USA) with primers listed in Appendix A. Cq values were used to calculate Fold Change differences between the control (TATA-box Binding Protein1, TBP1) and the treatments using the 2(-Delta Delta C(T)) method following [51] and calculating significant differences using Student’s *t*-test in Microsoft Excel.

### 5.8. Plant Metabolite Assays

Plant hormones (cytokinins, jasmonate, salicylic acid, auxin, *cis*-zeatin, and *trans*-zeatin) were measured by HPLC-mass spectrometry (HPLC-MS) as described previously [52]. B73 and *Hsf1* mutants were grown in a greenhouse to the V3 stage. One hundred mg of tissue was harvested from the distal end of the second true leaf of plants, immediately frozen in liquid N2, and stored in a −80 °C freezer until processing. For metabolite measurements across the growth zone, the ninth leaf was used. JA was extracted from tissue and quantified by LC-MS/MS (Appendix A). Then, 100 mg of tissue was mixed with 500 μL of phytohormone extraction buffer (1-propanol/water/HCl [2:1:0.002 vol/vol/vol]) containing 500 nM of d-JA (2,4,4-d3; acetyl-2,2-d2 JA (CDN Isotopes, Pointe-Claire, QC, Canada). The samples were shaken for 30 min in darkness, 500 μL of dichloromethane was added to each sample, and again agitated for an additional 30 min in darkness. The samples were then centrifuged at 14,000× *g* for 5 min and the lower organic layer of each sample was transferred to a glass vial for evaporation with nitrogen gas. Samples were then resuspended in 150 μL of methanol and syringe filtered through 0.20 μm polytetrafluoroethylene filters (Millipore, Burlington, MA, USA) to remove cellular debris. Samples were then placed in an insert inside a glass autosampler vial for analysis by liquid chromatography–tandem mass spectrometry (LC-MS/MS). For analysis, a Vanquish (Thermo, Waltham, MA, USA) high-performance liquid chromatography system was connected to a ZORBAX Eclipse Plus C18 column (Agilent, Santa Clara, CA, USA) and a TSQ Quantis mass spectrometer (Thermo, Waltham, MA, USA). Analytes were ionized by electrospray ionization and detected via multiple reaction mentoring. The injection volume was 3 μL and had a 450 μL min^−1^ mobile phase consisting of Solution A (0.2% acetic acid in water) and Solution B (0.2% acetic acid in acetonitrile) with a gradient consisting of (time—%B): 0 min—5%, 20 min—100%, 25.0—100%, 30 min—5%, —stop. All metabolites were quantitated relative to internal standards.

### 5.9. Statistics

Statistics were performed using R and R Studio [53] or Microsoft Excel. Statistical differences in final leaf measurements were calculated using a general linear model followed by Tukey’s HSD using the package ‘multcomp’ [54].

## Figures and Tables

**Figure 1 plants-12-03014-f001:**
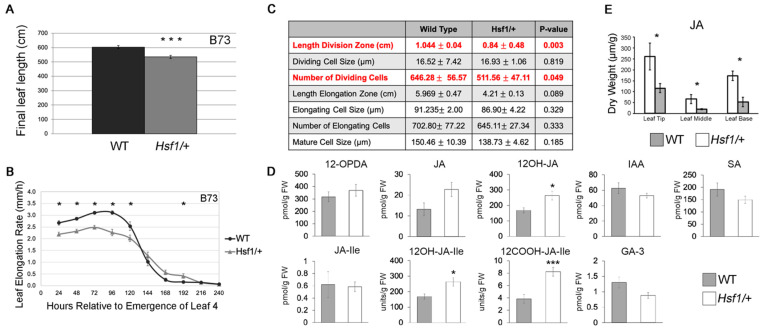
*Hsf1* growth and phytohormone phenotypes. (**A**) Barplots of WT and *Hsf1*/+ final leaf lengths. Error bars = SE. (**B**) Average leaf elongation rate (LER) of leaf #4 of *Hsf1*/+ and WT siblings in the B73 inbred background. Error bars = SE. (**C**) Kinematic analysis comparing growth zones of the *Hsf1*/+ mutant and its WT sibling. (**D**) Two-week-old whole-seedling hormone profile of *Hsf1*/+ and WT siblings. 12-OPDA, 12-oxophytodienoic acid; JA, Jasmonic Acid; 12OH-JA, 12-hydroxy-jasmonic acid; IAA, Indole-3-Acetic Acid; SA, Salicylic Acid; JA-Ile, Jasmonoyl Isoleucine; 12OH-JA-Ile, 12-hydroxy-jasmonoyl-isoleucine; 12COOH-JA-Ile, 12-carboxy-jasmonoyl-isoleucine; GA-3, gibberellic acid. (**E**) Jasmonic Acid (JA) concentration across leaf #9 at steady-state growth. The leaf was divided into three sections (leaf base, leaf middle, and leaf tip). Leaf base included the growth zone. White columns are *Hsf1*/+ and gray columns are WT sibling. Asterisks mark significant *p*-value differences (* *p* < 0.05, *** *p* < 0.001) calculated from a two-tailed Student’s *t*-test.

**Figure 2 plants-12-03014-f002:**
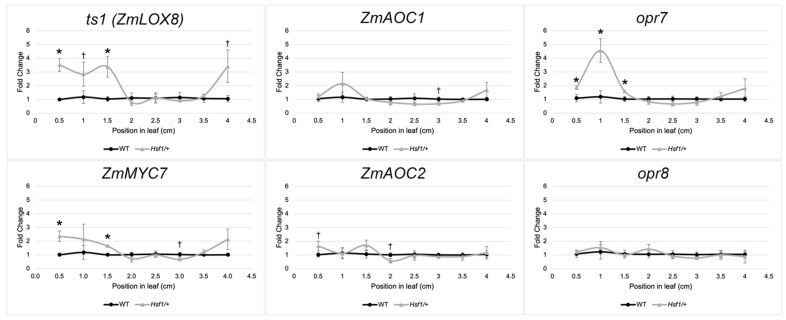
JA pathway genes are upregulated in the growth zone of *Hsf1* leaves. RT-qPCR of key JA biosynthesis and signaling genes across the division zone in *Hsf1/*+ and wild-type leaf #4 at steady-state growth. Asterisks and daggers mark significant differences by a one-tailed Student’s *t*-test. *p*-values * ≤ 0.05, † < 0.10.

**Figure 3 plants-12-03014-f003:**
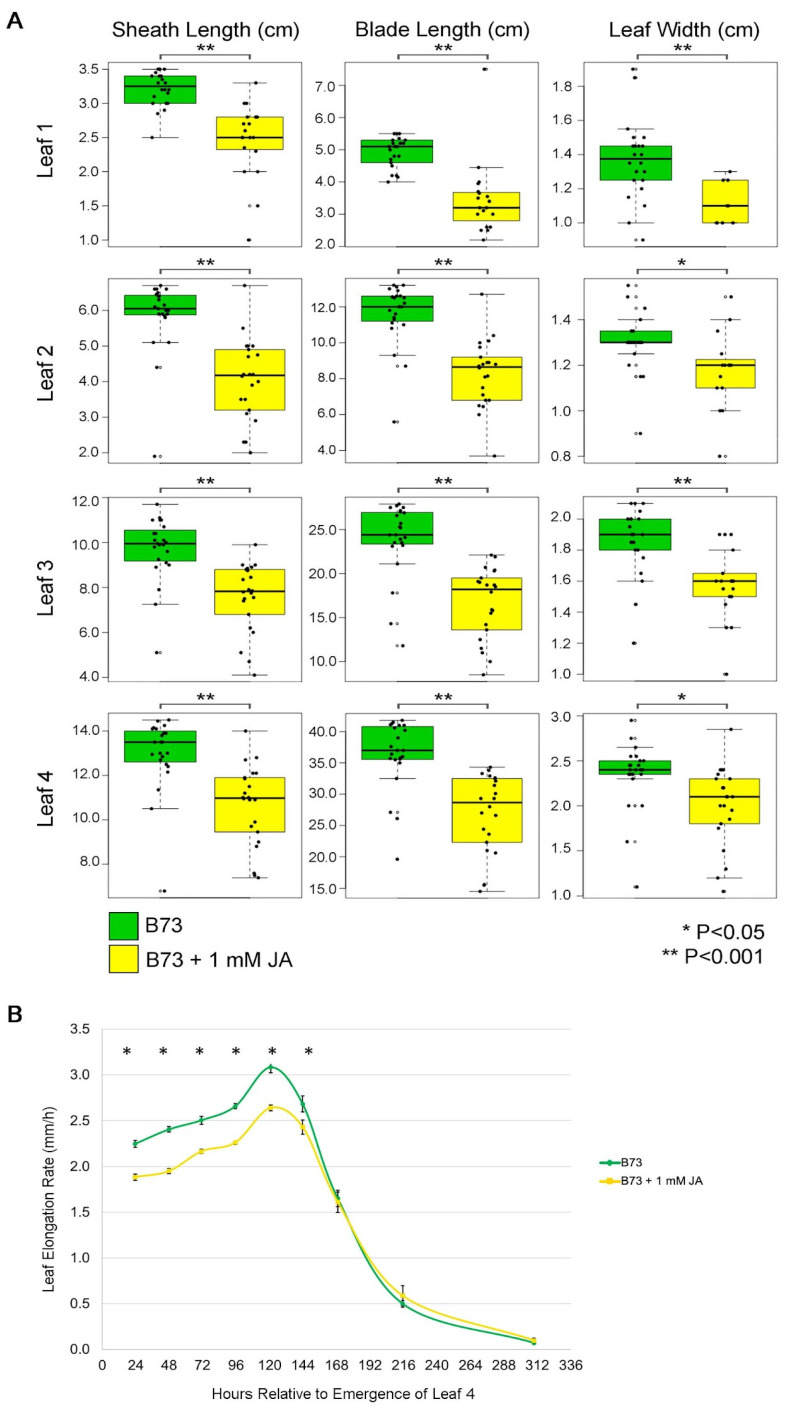
Effect of JA on B73 leaf growth. (**A**) Boxplots of sheath length, blade length, and blade width for leaves #1–#4 in control and 1 mM JA treated seedlings. Horizontal bars represent the maximum, third quantile, median, first quantile, and minimum values respectively. Each dot is a plant (B73, *n* = 23; B73 + JA, *n* = 22). Asterisks mark significant differences by Student’s *t*-test. (**B**) Average leaf elongation rate (LER) of leaf #4 at steady-state growth of control and 1 mM JA treated seedlings. Error bars = SE. Asterisks mark significant differences of LER between treatments at each time point by Student’s *t*-test *p*-value ≤ 0.05 (B73, *n* = 27; B73 + JA; *n* = 22).

**Figure 4 plants-12-03014-f004:**
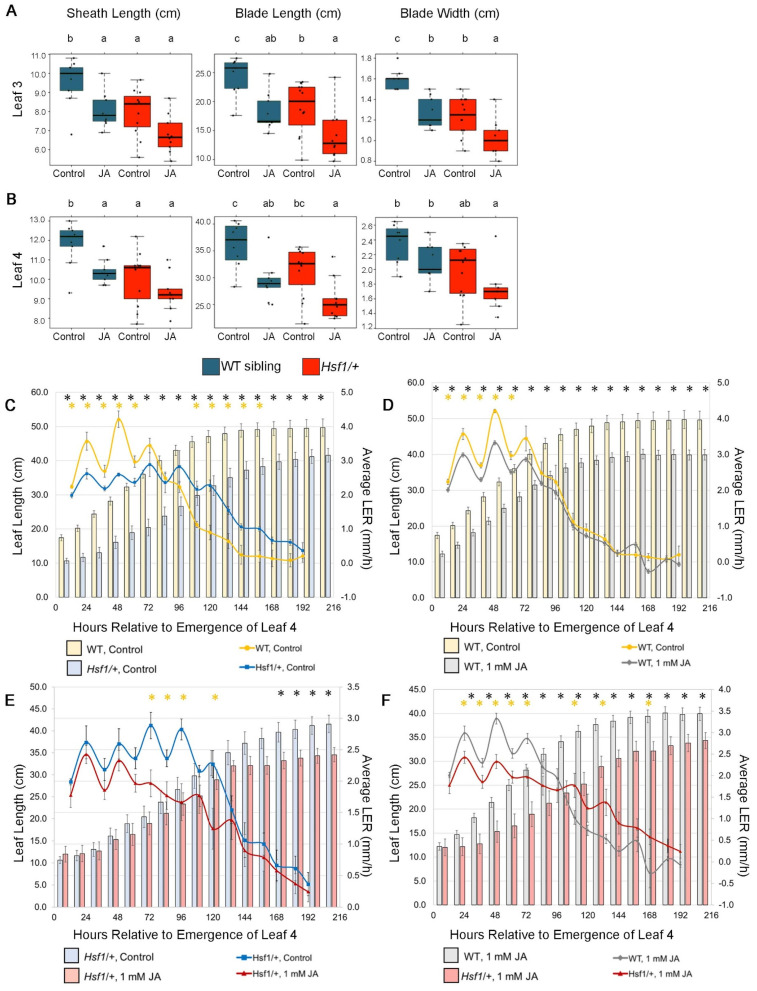
Final leaf size, leaf elongation rate (LER), and leaf elongation duration (LED) of *Hsf1*/+ and WT siblings treated with 1 mM JA. Boxplots of leaves #3 (**A**) and #4 (**B**) of *Hsf1*/+ and WT siblings from seedlings grown from germinating seed subjected to a 6-day, 1 mM JA treatment. Horizontal bars represent the maximum, third quantile, median, first quantile, and minimum values, respectively. Each dot is a plant (WT Control, *n* = 7; WT JA, *n* = 9; *Hsf1*/+ Control, *n* = 10; *Hsf1*/+ JA, *n* = 9). Letters unshared in compact letter display indicates significant differences by ANOVA followed by Tukey’s HSD. (**C**–**E**) LER superimposed over total leaf length. (**C**) LER and leaf lengths of WT and *Hsf1*/+ control treatments. JA treatment comparisons in (**D**) WT, (**E**) *Hsf1*/+, and (**F**) treated *Hsf1*/+ and WT. Significant differences by Student’s *t*-test are marked by asterisks. Yellow asterisks mark differences in LER and black asterisks mark differences in leaf length. Significant differences *p* < 0.05 are calculated by Student’s *t*-test. Error bars = SE.

**Figure 5 plants-12-03014-f005:**
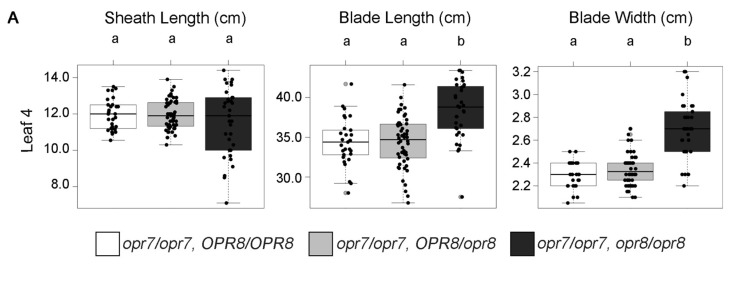
JA deficiency in maize enhances leaf growth. (**A**) Boxplots of sheath length, blade length, and blade width of the JA-deficient *opr7 opr8* double mutant as compared to its JA-sufficient siblings (*opr7/opr7*, *OPR8/OPR8* and *opr7/opr7*, *OPR8/opr8*). Unshared letters in compact letter display indicates significant differences by ANOVA followed by Tukey’s HSD. (**B**) LER of JA-deficient *opr7 opr8* double mutant as compared to its JA-sufficient siblings *opr7/opr7, OPR8/OPR8* and *opr7/opr7*, *OPR8/opr8*. Significant differences by Student’s *t*-test (*p* ≤ 0.05) for the double mutant compared to *opr7/opr7*, *OPR8/OPR8* or *opr7/opr7*, *OPR8/opr8* are indicated by black triangles or black squares, respectively. Error bars = SE (*OPR8/OPR8*, *n* = 34; *OPR8/opr8*, *n* = 62; *opr8/opr8*, *n* = 33). (**C**) LER of JA-deficient *Ts5* (*n* = 9) dominant mutant compared to its JA-sufficient WT sibling (*n* = 12). Error bars = SE. Asterisks mark significant differences by Student’s *t*-test *p*-value ≤ 0.05.

**Figure 6 plants-12-03014-f006:**
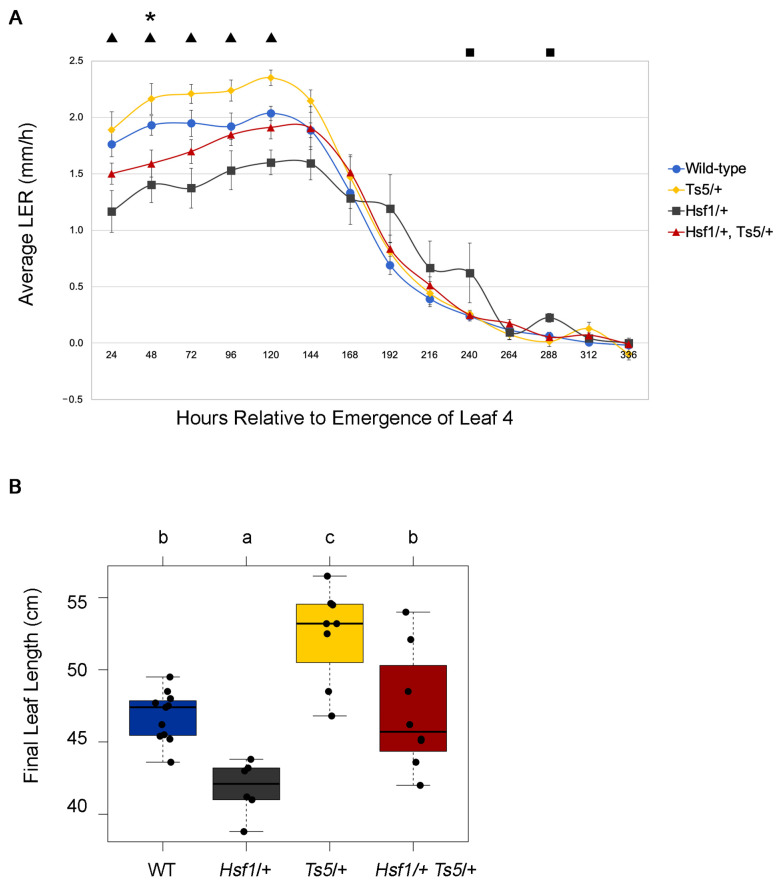
Epistatic interaction of *Hsf1* and *Ts5*. (**A**) LER of *Hsf1*/+ *Ts5*/+ double mutant compared to WT (blue circle, also Figure 5C), *Hsf1*/+ (black squares), and *Ts5*/+ (yellow diamond, also Figure 5C). Black asterisks, squares, or triangles above the LERs mark significant differences by Student’s *t*-test *p*-value ≤ 0.05 for double mutant compared to WT, *Hsf1*/+, or *Ts5/*+, respectively. Error bars = SE (+/+, *n* = 12; *Hsf1*/+, *n* = 6; *Ts5*/+, *n* = 9, *Hsf1*/+ *Ts5*/+, *n* = 10). (**B**) Boxplots of sheath length, blade length, and blade width of leaf #1 and #2 of the population described in (**A**) where leaf #1 is the leaf subtending the ear and leaf #2 is the next apical leaf. Horizontal bars represent the maximum, third quantile, median, first quantile, and minimum values, respectively. Each dot is a plant. Unshared letters in compact letter display indicates significant differences by ANOVA followed by Tukey’s HSD.

**Figure 7 plants-12-03014-f007:**
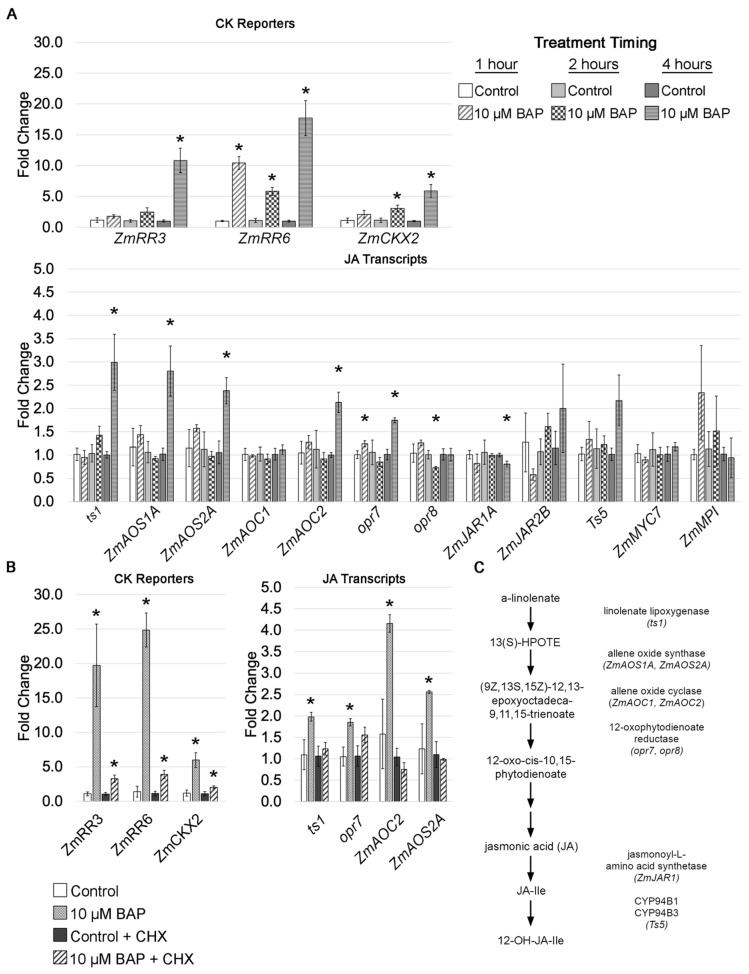
CK induces JA pathway gene expression in the leaf growth zone. (**A**) Quantitative real-time PCR analysis of CK reporter genes and JA biosynthesis and signaling genes after 10 µM BAP time course. (**B**) Quantitative real-time PCR analysis of CK reporter genes and JA biosynthesis genes after 10 µM BAP with and without cycloheximide (CHX) treatment. (**C**) Synopsis of JA pathway genes surveyed in (**A**,**B**). Asterisks in (**A**–**C**) mark significant differences (*p* < 0.05) between treatment and respective control calculated using a Student’s *t*-test.

## Data Availability

The data presented in this study are available in the Appendix A.

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
