# Peer review of "Cytokinin Promotes Jasmonic Acid Accumulation in the Control of Maize Leaf Growth"

_plants, 2023, doi:10.3390/plants12163014_

Round 1
Reviewer 1 Report
Plant organ growth is a dynamic process that occurs through the coordinated activities of cell division and cell expansion. These two processes are tightly regulated by a complex interplay of various hormones, which ultimately determine the final size of the organ. In this study, it was discovered that CK (cytokinin) has the ability to decrease leaf size and growth rate by suppressing cell division. Additionally, it was observed that the Hsf1 mutant, which exhibits hypersignaling of CK, showed elevated levels of jasmonic acid (JA), a hormone known to inhibit cell division. Exposing wild-type seedlings to exogenous JA (jasmonic acid) was found to decrease maize leaf size and growth rate. Conversely, maize mutants deficient in JA exhibited increased leaf size and growth rate. Analysis of gene expression indicated elevated levels of transcripts related to the JA pathway in the leaf growth zone of the Hsf1 mutant. Furthermore, when wild-type maize shoots were transiently treated with exogenous CK (cytokinin), it stimulated the expression of genes involved in the JA pathway. However, this effect was blocked when co-treated with cycloheximide. These findings suggest that CK has the potential to promote JA accumulation, possibly by enhancing the expression of specific genes involved in the JA pathway. Overall the manuscript is written well and logically laid out.
Author Response
Dear Plants Editor,
We thank the three Reviewers for their time and effort in reviewing our manuscript. To address their evaluations, we have made several changes including, adding new JA metabolite quantifications, redoing figures 1, 2, and 7, making clarifying edits to the narrative, and updating the cited references. All the edits to the manuscript and supplementary data files are shown as “track changes”. Below, we address each reviewers comment.
We thank the reviewer for their comments and hope the revisions made add to its clarity and logic.
Reviewer 2 Report
The study seems to be appropriate
The data is given in a well mannered fashion
The results and discussion are appropriate
good quality
Author Response
Dear Plants Editor,
We thank the three Reviewers for their time and effort in reviewing our manuscript. To address their evaluations, we have made several changes including, adding new JA metabolite quantifications, redoing figures 1, 2, and 7, making clarifying edits to the narrative, and updating the cited references. All the edits to the manuscript and supplementary data files are shown as “track changes”. Below, we address each reviewers comment.
We thank the reviewer for their comments and hope the revisions made have improved the paper.
Reviewer 3 Report
Since most of the JA biosynthesis pathway genes are up-regulated in Hsf1 mutant (Figure 2) and induced by CK (Figure 7) and only the end products (JA and JA-Ile) are determined (Figure 1), it is suggested to measure the intermediates such as 12-oxophytodienoic acid (OPDA), 13-hydroperoxylinolenic acid (13-HPOT) and other derivatives such methyl jasmonate (MeJA), which might be responsible for the to the Hsf1 growth phenotypes.
Classical defense hormones include SA, JA, and ethylene (ET), a large number of studies have shown that JA and ET act synergistically under certain circumstances. It would be better to measure ET and more phytohormone contents.
“Plant hormones are known to exert their function through crosstalk with other hormones (Santner and Estelle, 2009; De Vleesschauwer et al., 2014; Huot et al., 2014).” Please include more recent references (for example, doi: 10.3390/ijms22062914 and doi: 10.3390/ijms20030671).
One technological question on gene expression analysis: whether the expression of target genes in each sample was normalized to an endogenous reference (internal control gene) and what is the primer name/ sequence?
The manuscript seems to be prepared in a bit of a hurry with many typos/grammatical mistakes. Authors need to check and edit their work thoroughly.
Author Response
Dear Plants Editor,
We thank the three Reviewers for their time and effort in reviewing our manuscript. To address their evaluations, we have made several changes including, adding new JA metabolite quantifications, redoing figures 1, 2, and 7, making clarifying edits to the narrative, and updating the cited references. All the edits to the manuscript and supplementary data files are shown as “track changes”. Below, we address each reviewers comment.
Comments and Suggestions for Authors
Since most of the JA biosynthesis pathway genes are up-regulated in Hsf1 mutant (Figure 2) and induced by CK (Figure 7) and only the end products (JA and JA-Ile) are determined (Figure 1), it is suggested to measure the intermediates such as 12-oxophytodienoic acid (OPDA), 13-hydroperoxylinolenic acid (13-HPOT) and other derivatives such methyl jasmonate (MeJA), which might be responsible for the to the Hsf1 growth phenotypes.
We thank the reviewer for their comment. Regarding JA derivatives, there are dozens of derivatives that exist, but only a few (including methyl jasmonate, which would require an entirely different type of analysis) have confirmed biological function. To address these concerns, we have added accumulation data for several types of jasmonates that can serve as precursor/derivatives of biologically active jasmonates. We have also included analysis of 12-OPDA, a precursor of JA, which shows little to no induction. With the addition of these new metabolite data, we think that measuring the precursor of 12-OPDA, 13S-HPOT, which is difficult to measure due to its inherent instability and quick degradation, is not needed.
Classical defense hormones include SA, JA, and ethylene (ET), a large number of studies have shown that JA and ET act synergistically under certain circumstances. It would be better to measure ET and more phytohormone contents.
Thank you for the comment. There is indeed a substantial amount of plant hormone cross-talk at all levels of regulation (transcriptional, translational, protein, metabolic, etc.). However ethylene is not trivial to measure, and would require an entirely separate analysis. To help address the reviewer’s comments about potential hormone-crosstalk, we have added the analysis of the major phytohormones, salicylic acid (SA), gibberlic acid (GA), and indole-3-acetic acid (IAA). Measuring additional phytohormone content, although of value, is beyond the scope of this study, which focuses on the connection between CK and JA.
“Plant hormones are known to exert their function through crosstalk with other hormones (Santner and Estelle, 2009; De Vleesschauwer et al., 2014; Huot et al., 2014).” Please include more recent references (for example, doi: 10.3390/ijms22062914 and doi: 10.3390/ijms20030671).
Thank you for your helpful suggestion. We have updated the relevant references.
One technological question on gene expression analysis: whether the expression of target genes in each sample was normalized to an endogenous reference (internal control gene) and what is the primer name/ sequence?
Thank you for pointing this out. The housekeeping gene information was mistakenly left off the Suppl. Table 5, which has now been fixed. We also added more details on how the transcript fold-change was determined in the gene expression section in Methods.
The manuscript seems to be prepared in a bit of a hurry with many typos/grammatical mistakes. Authors need to check and edit their work thoroughly.
Thank you for the critical feedback. Additional edits were made to improve the clarity and flow of the manuscript.
Round 2
Reviewer 3 Report
The authors has made considerable efforts to improve the manuscript and it is acceptable for publication now.